# Acute Kidney Injury after Endoscopic Retrograde Cholangiopancreatography—A Hospital-Based Prospective Observational Study

**DOI:** 10.3390/biomedicines10123166

**Published:** 2022-12-07

**Authors:** Florica Gadalean, Florina Parv, Oana Milas, Ligia Petrica, Iulia Ratiu, Bogdan Miutescu, Adrian Goldis, Cristina Gluhovschi, Flaviu Bob, Anca Simulescu, Mihaela Patruica, Adrian Apostol, Viviana Ivan, Adalbert Schiller, Daniela Radu

**Affiliations:** 1Department of Internal Medicine II—Division of Nephrology, “Victor Babeș” University of Medicine and Pharmacy Timisoara, Eftimie Murgu Sq. No. 2, 300041 Timisoara, Romania; 2County Emergency Hospital Timisoara, 300723 Timisoara, Romania; 3Centre for Molecular Research in Nephrology and Vascular Disease, Faculty of Medicine, “Victor Babeș” University of Medicine and Pharmacy, Eftimie Murgu Sq. No. 2, 300041 Timisoara, Romania; 4Department of Internal Medicine II—Division of Cardiology, “Victor Babeș” University of Medicine and Pharmacy Timisoara, Eftimie Murgu Sq. No. 2, 300041 Timisoara, Romania; 5Center for Translational Research and Systems Medicine, Faculty of Medicine, “Victor Babeș” University of Medicine and Pharmacy, Eftimie Murgu Sq. No. 2, 300041 Timisoara, Romania; 6Centre for Cognitive Research in Neuropsychiatric Pathology (Neuropsy-Cog), Faculty of Medicine, “Victor Babeș” University of Medicine and Pharmacy, Eftimie Murgu Sq. No. 2, 300041 Timisoara, Romania; 7Department of Internal Medicine II—Division of Gastroenterology, “Victor Babeș” University of Medicine and Pharmacy Timisoara, Eftimie Murgu Sq. No. 2, 300041 Timisoara, Romania; 8Department X Surgery II—Division of Surgery I, “Victor Babeș” University of Medicine and Pharmacy Timisoara, Eftimie Murgu Sq. No. 2, 300041 Timisoara, Romania

**Keywords:** acute kidney injury, endoscopic retrograde cholangiopancreatography, in-hospital mortality, risk factors

## Abstract

Background: Endoscopic retrograde cholangiopancreatography (ERCP) represents a major pivotal point in gastrointestinal endoscopy. Little is known about acute kidney injury (AKI) post-ERCP. This study analyses the incidence, risk factors, and prognosis of post-ERCP AKI. Methods: A total of 396 patients were prospectively studied. AKI was defined by an increase in serum creatinine (SCr) ≥ 0.3 mg/dL or by an increase in SCr ≥ 50% in the first 48 h post-ERCP. Logistic regression analysis was used to identify the predictors of AKI and in-hospital mortality. A two-tailed *p* value < 0.05 was considered significant. Results: One hundred and three patients (26%) developed post-ERCP AKI. Estimated glomerular filtration rate (adjusted odds ratio (aOR) = 0.95, 95% confidence interval (CI): 0.94–0.96, *p* < 0.001), nonrenal Charlson Comorbidity Index (Aor = 1.19, 95% CI: 1.05–1.35, *p* = 0.006), choledocholithiasis (aOR = 4.05, 95% CI: 1.98–8.29, *p* < 0.001), and bilirubin (aOR = 1.1, 95% CI: 1.05–1.15, *p* < 0.001) were associated with post-ERCP AKI. Post-ERCP AKI was associated with longer hospital stay (*p* < 0.001) and with increased in-hospital mortality (7.76% versus 0.36%, *p* < 0.001). Moderate-to-severe (stage 2 and 3) AKI was independently associated with in-hospital mortality (aOR = 6.43, 95% CI: 1.48–27.88, *p* < 0.013). Conclusions: Post-ERCP AKI represented an important complication associated with longer hospital stay. Moderate-to-severe post-ERCP AKI was an independent risk factor for in-hospital mortality.

## 1. Introduction

Endoscopic retrograde cholangiopancreatography (ERCP) is considered a great innovation related to the management of individuals with pancreaticobiliary diseases, although it is a complex and technically challenging procedure that implies the highest risk for complications among all routine endoscopic procedures [1]. Around the globe, ERCP volumes have continued to rise over the past 10–15 years [2]. The use of therapeutic ERCP has increased 30-fold in recent decades [3], with approximately 600,000 procedures performed in the United States and 1,3 million worldwide each year [1].

Acute kidney injury (AKI) is a severe clinical syndrome reported in patients undergoing endoscopic procedures such as colonoscopy [4] or endoscopy of the upper gastrointestinal tract for variceal [5] and nonvariceal bleeding [6,7]. Moreover, AKI occurs early after endoscopy and is associated with longer hospital stays, increased costs, and both short- and long-term mortality [5,6,7].

The incidence of AKI in patients who underwent ERCP has been poorly described in the literature. Chronic kidney disease (CKD) and end-stage renal disease (ESRD) have a known association with post-ERCP prolonged hospital stays, higher in-hospital mortality rates, and considerably larger hospital charges [8,9]. Furthermore, ESRD and CKD are associated with higher post-ERCP adverse events, including bleeding and post-ERCP pancreatitis [9,10]. By contrast, there are scarce data available concerning AKI post-ERCP and its potential impact on patient’s outcome. Few studies have shown a prevalence of AKI post-ERCP ranging from 11.48% to 17% [11,12,13]. It is assumed that the occurrence of AKI after ERCP is an independent risk factor for in-hospital mortality [14].

Due to a tremendous lack of data on AKI following ERCP, the aim of the present study was to investigate the incidence of AKI after ERCP in a prospective representative cohort undergoing ERCP procedures in a tertiary teaching hospital. In addition, potential independent factors associated with the occurrence of AKI post-ERCP and with in-hospital mortality were assessed in these patients.

## 2. Materials and Methods

In this prospective study were included patients aged 18 years or above, admitted to the Department of Gastroenterology of the County Hospital Timisoara, Romania, between January 2019 and February 2020, who underwent scheduled or urgent ERCP. The County Emergency Hospital of Timisoara is a tertiary and teaching hospital providing medical assistance to the Western Region of Romania with almost 2,000,000 inhabitants.

Exclusion criteria were CKD on renal replacement therapy, requirement for renal replacement therapy in the week before ERCP, less than 48 h of hospital stay, and/or fewer than two assessments of serum creatinine (SCr). For patients undergoing more than one ERCP session within the same hospitalization, only the first ERCP was considered for analysis.

### 2.1. Ethics Statement

The County Emergency Hospital of Timisoara Ethical Committee (Board of Human Studies) approved the protocol (approval number 2/3 January 2019), and every patient provided written informed consent before enrolment. The study was performed and reported in conformity with Strengthening the Reporting of Observational Studies in Epidemiology (STROBE) recommendations [15].

### 2.2. Variables

Data concerning demographics and comorbidities were retrieved from the GP’s files. Comorbidity was defined as a pre-existing illness present upon admission to the hospital. Medical conditions, including hypertension, myocardial infarction, congestive heart failure, cerebrovascular disease, peripheral vascular disease, diabetes mellitus, pre-existing kidney disease, peptic ulcer disease, chronic liver disease, dementia, chronic pulmonary disease, connective tissue disorders or autoimmune disease, malignancies (solid and haematologic) with or without metastases, and acquired immunodeficiency syndrome, were recorded. Comorbidities were then accurately accounted using the age-adjusted Charlson Comorbidity Index (CCI) score because of its enhanced predictive accuracy according to the second edition of CCI [16].

Initial laboratory results, including complete blood count, glucose, alkaline phosphatase (ALP), alanine aminotransferase (ALT), aspartate aminotransferase (AST), bilirubin, lipase, serum albumin, C-reactive protein (CRP), and renal function (creatinine levels), were recorded. Serum creatinine values were measured on admission, followed with routinely evaluations (usually daily) during hospitalization. The systemic inflammatory response syndrome (SIRS) at the time of admission was defined as the presence of at least 2 of the following criteria: temperature > 38 °C or <36 °C, pulse > 90 beats/min, respiratory rate > 20 breaths/min or PaCO_2_ < 32 mmHg, or white blood cell count < 4000 cells/mm^3^ or >12,000 cells/mm^3^ or >10% immature bands. [17]. Preprocedural acute cholangitis (AC) was defined according to the 2018 Tokyo Guidelines, and the diagnostic criteria included systemic inflammation, cholestasis, and imaging findings [18].

### 2.3. Estimation of Renal Function and Definition of AKI

Baseline renal function was assessed using the CKD-EPI creatinine equation [19]. Serum creatinine at admission was considered to be the baseline. AKI was defined and staged according to the Kidney Disease Improving Global Outcomes (KDGIO) criteria. Therefore, AKI was defined as an increase in serum creatinine (SCr) by ≥0.3 mg/dL (≥26.5 μmol/L) or a rise in SCr to ≥1.5 times the baseline value. The staging of AKI according to severity was evaluated based on the magnitude of changes in SCr: stage 1 if there was an increase in absolute SCr of at least 0.3 mg/dL or a SCr increase to 1.5–1.9 times baseline value; stage 2 if there was a SCr increase to 2–2.9 times baseline value; and stage 3 if there was a SCr increase to ≥3 times the baseline value or a peak SCr > 4 mg/dL, with at least a 0.3 mg/dL increase within the 48 h time window [20]. AKI was only considered within the first 48 h after ERCP in order to relate AKI to the ERCP procedure itself, thus eliminating the potential interaction with other factors occurring in the post-ERCP period. None of the patients required dialysis.

### 2.4. Endoscopic Retrograde Cholangiopancreatography

Endoscopic retrograde cholangiopancreatography was performed with an Olympus TJF-145 Video Duodenoscope (Olympus Medical Systems, Tokyo, Japan). In our department, all patients undergoing ERCP received periprocedural hydration with normal intravenous saline on a routine basis, with a maximum of 1.5 mL/kg per h and 3 L per 24 h, and the hydration measures were adapted to the patient’s comorbidities (i.e., congestive heart failure) in order to avoid overhydration-related complications, such as pulmonary edema and congestive heart failure. The procedures were performed safely while the patient was under sedation or general anaesthesia according to the anaesthesiologist indications. All included patients provided informed consent before the ERCP procedure.

### 2.5. Outcomes

The following two clinical outcomes were assessed: (1) incidence and risk factors of AKI development and (2) in-hospital mortality rate associated to AKI in the first 48 h post-ERCP.

### 2.6. Statistical Analysis

The analysis of data was performed using the SPSS v.17 software suite (SPSS Inc. Chicago, IL, USA), and data were presented as mean ± standard deviations for continuous variables with Gaussian distribution, median (interquartile range (IQR)) for continuous variables without Gaussian distribution, or percentages for categorical variables. Continuous variables distributions were checked for optimal functioning using the Shapiro–Wilk test, and for variance equality using Levene’s test. To evaluate the significant differences between groups, the Student’s *t*-test (means, Gaussian populations), Mann–Whitney-U test (medians, non-Gaussian populations), and chi-square (proportions) were utilized.

Logistic regression analysis was used to assess independent predictors of AKI and in-hospital mortality. Univariate logistic regression analysis was initially performed to assess which parameters at admission were associated with AKI development and with in-hospital mortality, respectively. Then, the variables that were statistically significant (*p* < 0.05) in the univariate analysis were included in the multivariate analysis using time-dependent Cox regression model to identify independent factors associated with AKI development and in-hospital mortality. The continuous variables were registered in the model as variables of continuity and categorical variables as categorical ones. Given that CCI score includes two points for CKD, we excluded CKD from CCI and calculated a nonrenal CCI score, which was used as covariate in the adjusted analysis to control for multicollinearity with SCr and AKI. Moreover, components of the SIRS were excluded in the multivariate analyses in order to avoid multicollinearity. Data were presented as odds ratios (ORs) with 95% confidence intervals (95% CI).

Survivability was evaluated by Kaplan–Meier survival curves and compared using the log-rank test. A two-sided *p* value of <0.05 was considered as the threshold for statistical significance.

## 3. Results

During the study period, 438 patients underwent ERCP procedures in the Gastroenterology Department of County Emergency Hospital of Timisoara; three of them displayed end-stage renal disease on dialysis and were not included. None of the patients had received renal transplants. Of the 435 remaining patients, 39 individuals were hospitalized in less than 48 h and/or had fewer than two SCr determinations during the hospitalization, and were excluded from the study. Therefore, in this analysis, 396 patients were included and evaluated as a cohort. Baseline and clinical characteristics are shown in Table 1. The median age of the studied patients was 69 years. Male gender accounted for 46.2% of cases. The most common indication for ERCP was choledocholithiasis (60.6%), followed by malignant biliary obstruction (21.2%), and biliary stricture (18.18%). The most utilized therapeutic procedure was biliary or pancreatic drainage (68.4%), followed by sphincterotomy (51.5%). Biopsy was reported for 21 cases (5.3%). In our study, duration of ERCP procedure ranged between 10 and 110 min, with a mean procedure duration of 34.9 ± 9.7 min (Table 1).

### 3.1. Incidence and Risk Factors of AKI

One hundred and three patients (26%) developed AKI in the first 48 h after ERCP, as follows: 78 patients (19.7%) remained at stage 1, 13 patients (3.3%) were at stage 2, and 12 patients (3%) ended up at stage 3. Patients with post-ERCP AKI were older (*p* = 0.004) and more likely to have SIRS upon admission time (*p* = 0.018) with higher CCI score (*p* < 0.001). At admission time, sepsis was more prevalent in the group with AKI versus non-AKI patients (*p* < 0.001). Furthermore, upon admission, patients with AKI were more likely to have higher white blood cell counts (*p* = 0.001) and neutrophil counts (*p* = 0.0004), higher levels of CRP (*p* < 0.001), lower levels of haematocrit (*p* = 0.013) platelet counts (*p* = 0.002), and serum albumin (*p* < 0.001), and lower values of baseline eGFR (*p* < 0.001), respectively. In addition, individuals with AKI after ERCP had significantly higher levels of bilirubin (*p* < 0.001), ALP (*p* < 0.001), and glycaemia (*p* = 0.010) (Table 1). Choledocholithiasis was more prevalent among patients with post-ERCP AKI than in patients who did not have AKI (*p* = 0.026), whereas biliary/pancreatic duct stricture was far more frequent in patients without post-ERCP AKI (*p* = 0.001). Preprocedural acute cholangitis, defined according to the 2018 Tokyo Guidelines, was found in 56.31% of patients, the rate of AC being significantly higher in the AKI group versus non-AKI patients (79.61% vs. 48.12%; *p* < 0.001). (Table 1). Causes of AC included choledocholithiasis (63.67%), followed by malignant biliary obstruction (20.62%), and biliary/pancreatic duct strictures (15.69%). Concerning the duration of ERCP procedure, there were no differences between the AKI versus no AKI group. (Table 1).

Univariate analysis showed multiple potential risk factors associated with AKI (Table 2).

All parameters exhibiting a univariate association with AKI were included in the multivariate analysis. Using the backward method, we identified the final model, which justified 85% of the AKI development (Nagelkerke R2 = 0.850), with the following parameters as independent predictors of post-ERCP AKI: baseline eGFR (OR = 0.91; 95% CI, 0.89–0.97), age (OR = 1.07; 95% CI, 1.03–1.12), non-renal CCI score (OR = 1.21; 95% CI, 1.02–1.44), choledocholithiasis (OR = 3.31; 95% CI, 1.29–8.54), and bilirubin (OR = 1.12; 95% CI, 1.06–1.18), (Table 2).

### 3.2. Outcome

Patients who developed AKI after ERCP experienced a significantly longer period of in-hospital stay, with a median of 7 days versus 5 days in the non-AKI group (*p* = 0.002). There were no significant differences between the AKI and non-AKI group concerning post-ERCP complications, such as pancreatitis, bleeding, and perforation (Table 3).

In our cohort, in-hospital mortality rate was 2.27%. Of the nine patients who died, progression of malignancy (3/9; 33.33%) and infections (3/9; 33.33%) were the major causes of death. All other causes of death are presented in Table 2. The occurrence of AKI after ERCP was associated with an increased incidence of in-hospital mortality when compared to the non-AKI group (7.76% versus 0.34%, *p* < 0.001). In the AKI group, deaths associated with the progression of underlying malignancies were more frequent as compared to non-AKI patients (*p* = 0.016). (Table 2) Risk of in-hospital death proportionally increased with severity of AKI, with the rate of mortality being 5.12% in patients with mild AKI (stage 1) and 16% in patients with moderate-to-severe AKI (stages 2 and 3), respectively.

Univariate analysis revealed that baseline eGFR (*p* = 0.022), serum albumin (*p* = 0.001), bilirubin (*p* = 0.015), CCI score (*p* = 0.001), SIRS (*p* < 0.026), neutrophils (*p* < 0.001), leucocytes (*p* < 0.001), and moderate-to-severe AKI (stages 2 and 3) (*p* < 0.001) were all associated with increased in-hospital mortality. Conversely, mild AKI (stage 1) (unadjusted OR 3.37, 95% CI 0.89–27.88, *p* = 0.074) was not associated with increased in-hospital mortality.

Furthermore, by using multivariate logistic regression, moderate-to-severe AKI (stages 2 and 3), lower serum albumin levels, and elevated non-renal CCI score were identified as independent risk factors of in-hospital mortality (Table 4).

Kaplan–Meier analysis revealed that the cumulative survival rate decreased concomitantly with the increase in AKI severity (*p* < 0.001). This discovery emphasizes worse short-term survival related to moderate-to-severe AKI in patients who underwent ERCP. The survival analysis and the time-series change of subject numbers are presented in Figure 1.

## 4. Discussion

In the present study, we prospectively analysed the incidence, predictors, and impact on in-hospital mortality of AKI in a cohort of 396 patients undergoing ERCP. The incidence of AKI reached an astonishing value of 26%, of which mild AKI (stage1) occurred in 19.7% of patients, while moderate-to-severe AKI (stage 2 and stage 3) developed in 6.3% of cases. The independent factors associated with AKI were baseline eGFR, nonrenal CCI score, choledocholithiasis, and serum bilirubin level at admission. Patients who developed AKI after ERCP presented a significantly longer period of in-hospital stay than non-AKI patients. Patients with AKI had an in-hospital mortality rate of 7.76%, which was substantially higher as compared to the non-AKI group. In our patients, in-hospital mortality rate gradually increased with the rising of post-ERCP AKI severity. Furthermore, in multivariate analysis, moderate-to-severe AKI (stages 2 and 3) was a strong independent predictor for in-hospital mortality, while AKI stage 1 did not present a linking point to short-term mortality.

Worldwide, AKI is a common complication in hospitalized patients, occurring in 20.0–31.7% of patients at diverse levels of in-hospital care [21], and being associated with significantly higher morbidity and short- and long-term mortality [22].

The incidence of AKI related to digestive endoscopic procedures, such as oesophagogastroduodenoscopy, varied from 6.37% to 48.7%, depending on many different factors such as AKI definitions, patient population, or indications for oesophagogastroduodenoscopy [5,6,7]. In the case of colonoscopy, AKI occurred soon after endoscopy, and the reported incidence of AKI ranged from 1.16% to 41.2% [5,23]. In contrast, the literature with regard to the incidence of AKI after ERCP exists in scarce amounts, despite the fact that nowadays ERCP is a common procedure used for treating diseases of the biliary and pancreatic ducts, and its utilization has increased 30-fold in the last decades [3].

Two previous retrospective studies reported an incidence of post-ERCP AKI around 17% [12,13]. Moreover, recent data published as conference abstract revealed in one large cohort of 1,942,313 ERCP procedures that incidence of AKI after ERCP was 11.48% [11]. In addition, during the study period, the authors observed a rise in post-ERCP AKI incidence from 6.4% in 2007 to 16.4% in 2018 [11]. In our cohort, the incidence of post-ERCP AKI had a rate of 26%, as compared to the data mentioned above. The higher rate of AKI in our cohort may be due to improved recognition of AKI, as we used the most sensitive definition of early AKI detection according to KDIGO. Moreover, in our cohort, there is an increased likelihood of including a large proportion of high-risk cases given the tertiary level of care of our hospital.

The pathophysiology of post-ERCP AKI is complex and only partially understood. ERCP itself triggers a systemic inflammatory response [24,25], which plays an important role in renal damage associated with AKI, leading to microvasculature dysfunction and alteration in tubular cells’ functions [26].

In the present study, we identified the following parameters as independent predictors for AKI: lower eGFR level, higher nonrenal CCI score, choledocholithiasis as indication for ERCP, and elevated levels of total bilirubin. Several large-scale studies have established that lower baseline eGFR is a reliable risk factor for AKI [27,28]. In our study, we observed that a higher CCI score represents an independent predictor for AKI. Currently, the CCI score is the most widely used instrument to assess the consequences of comorbid diseases on individual prognosis, and its use has been proven in kidney diseases, including AKI [29]. Thus, in a recent large cohort of 786 patients diagnosed with AKI, a CCI score greater than 6 was an independent predictor for poor renal outcomes [30]. In our works, we found that higher bilirubin levels upon admission were an independent predictor for AKI. Recently, a large retrospective study including a total amount of 9496 patients showed that total bilirubin > 2.0 mg/dL was an independent risk factor for contrast-induced acute kidney injury [31]. Furthermore, the association between hyperbilirubinemia and increased risk for AKI in patients with liver disease is seemingly a common finding [32]. The choledocholithiasis was independently associated with a higher risk of post-ERCP AKI. There is evidence of an independent relation between choledocholithiasis and atherosclerosis [33]. Atherosclerosis, which impairs kidney microcirculation [34], contributed to an increased risk of AKI, which was particularly observed after cardiac surgery [35].

In our study, patients with post-ERCP AKI had a significantly longer hospital stay compared to the non-AKI group. This observation is consistent with previous studies generally concerning AKI [36,37], and particularly in AKI, following digestive endoscopy for upper-gastrointestinal bleeding [6,7].

Previously, large-scale studies have demonstrated that in-hospital mortality associated with AKI is higher compared to a population without AKI [36,37,38]. In this study, in-hospital mortality associated with AKI was more than 20 times that of the non-AKI population, and worsened with severity of kidney injury. In our patients with post-ERCP AKI, the rate of in-hospital mortality was 7.76% compared with 0.3% in the non-AKI group. This finding is similar to the discovery in a large ERCP procedures database from the United States, in which post-ERCP AKI was associated with an in-hospital mortality rate of 7% as compared to 0.6% in ERCP patients without AKI [21].

In our study, we observed an independent association between AKI severity and in-hospital mortality, even after adjustment for available confounders in a Cox multivariate model. The moderate-to-severe AKI (stage 2 and stage 3) was an independent predictor for short-term mortality, with an odds ratio of in-hospital death more six-fold higher than patients with no AKI. This is consistent with the previous data concerning the correlation between AKI stages and risk of in-hospital mortality and which demonstrated that short-term mortality rate proportionally increases with the severity of AKI stage [36,37,38,39].

Although moderate-to-severe AKI has been found as an independent risk factor of in-hospital mortality, it is debatable if moderate-to-severe AKI is a cause of mortality per se or whether AKI is a consequence of worsening severity of underlying illness, thus leading to an excess of morbidity and mortality. The mechanism by which AKI could contribute to increased mortality remains incompletely understood, but the cross-talks between injured kidney and many distant systems/organs, such as the immune system, heart, brain, liver, gut, lung, and other vital organs are possible explanations [40]. The pathways of cross-talk between kidney and remote organs are multifactorial with the generation of proinflammatory mediators, activation of leucocyte and oxidative stress, endothelial dysfunction, channel dysregulation, cellular apoptosis, and neuroendocrine activation [40,41]. The complex crosstalk between kidney and distant organs induces systemic and organ-specific immunologic, humoral, and haemodynamic imbalances leading to major organ dysfunction, which contributes to increasing AKI-related morbidity and mortality [22,40,41]. There seems to be a higher probability of distant organ dysfunction with more severe AKI [41]. The observational nature of our study does not allow for conclusions regarding a direct causal relation between AKI and the excess mortality in subjects who underwent ERCP. For this reason, further studies are imperative to investigate causality between AKI and mortality in patients undergoing ERCP.

This study has several limitations. First, the single-centre design of the study reduces its potential of generalization. The results should be validated in a larger multicentre ERCP population. Second, measurements of urine output were not available in all patients, and therefore it is possible to have underestimated AKI events. Third, preadmission creatinine values were not available in all patients. The SCr level upon admission was defined as the baseline. There is a probability that upon admission, SCr had been already increased, leaving a risk of misestimating the total (pre- and post-ERCP) development of AKI. However, the aim of the study was to investigate the effect of the ERCP procedures on the risk of AKI. Therefore, the difference in SCr pre- versus post-ERCP was the primary objective of interest. Patients were defined as having post-ERCP AKI if they met the KDIGO criteria postprocedurally, irrespective of whether the kidney injury started before or after ERCP. Fourth, we were not able to delineate the mechanisms by which post-ERCP AKI could cause death, given the fact that our study had an observational nature.

However, our study possesses certain important strengths. To the best of our knowledge, for the first time the study describes the incidence, risk factors, and impact on patient’s outcome of AKI following ERCP. Moreover, we used the most standardized definition of AKI as stated by expert recommendations. In addition, many covariates with impact on the occurrence and outcome of AKI were investigated.

The results of our study highlight the fact that patients undergoing ERCP are vulnerable to development of AKI. For these patients, it is very important to intervene early and to provide an adequate periprocedural hydration, while avoiding fluid overload, and also to withdraw or not to add possible nephrotoxins.

## 5. Conclusions

In conclusion, in this observational prospective study, AKI was observed in 26% of patients undergoing ERCP. Furthermore, we found that baseline eGFR, nonrenal CCI score, choledocholithiasis, and bilirubin level may predict the development of AKI after ERCP. Patients with post-ERCP AKI experienced a prolonged hospital stay and higher rate of in-hospital mortality, but only moderate-to-severe AKI was an independent risk factor for in-hospital death. Our results underline the fact that it is of great importance for clinicians to conceive strategies for AKI screening, design appropriate risk-stratification instruments, and institute preventative measures to avoid the harmful role of AKI and its consequences among susceptible patients undergoing ERCP.

## Figures and Tables

**Figure 1 biomedicines-10-03166-f001:**
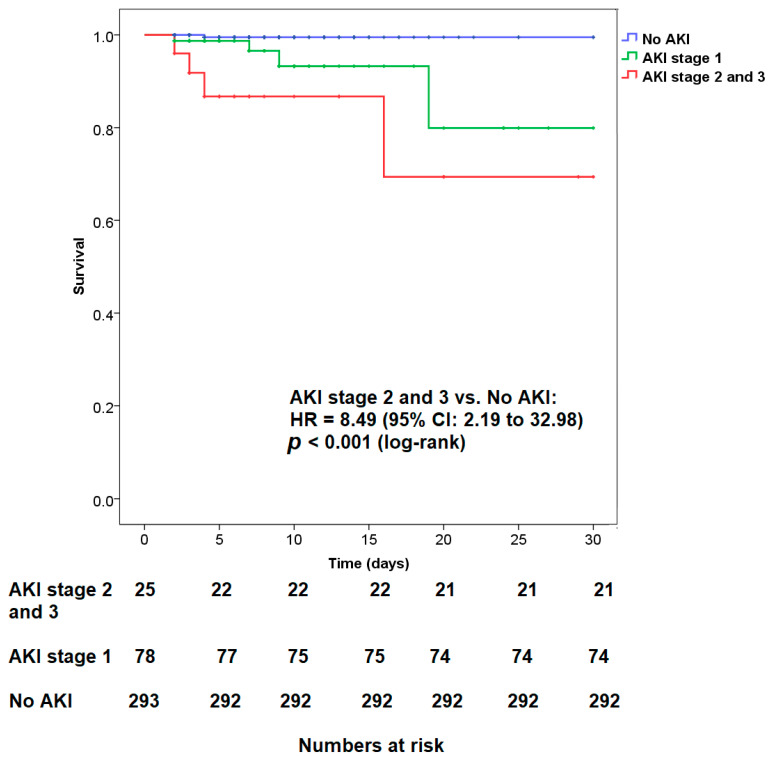
Cumulative survival rate of patients with mild post-ERCP AKI (stage 1) and moderate-to-severe post-ERCP AKI (AKI stages 2 and 3) during post-ERCP hospital stay.

**Table 1 biomedicines-10-03166-t001:** Baseline characteristics of the patients who underwent endoscopic retrograde cholangiopancreatography and comparison between patients with AKI and with no AKI after-ERCP.

Parameter		Total ERCP Group (*n* = 396)	AKI Group (*n* = 103)	Non-AKI Group (*n* = 293)	*p*
Age, (years) ^a^		69 {17}	70.83 ± 12.26	68 {18}	0.004
Male gender, *n* (%) ^c^		183 (46.21)	53 (51.45)	130 (44.36)	0.251
Clinical and laboratory data on admission					
	SIRS, *n* (%) ^c^	88 (22.22)	32 (31.06)	56 (19.11)	0.018
	CCI ^a^	5 {4}	6 {4}	5 {4}	<0.001
	Nonrenal CCI ^a^	5 {4}	6 {4}	5 {4}	0.001
	White blood cell counts (×10^3^/µL) ^a^	8.15 {4.2}	9.1 {5}	7.9 {3.9}	0.001
	Neutrophil counts (×10^3^/µL) ^a^	6.2 {4.4}	7.4 {5.2}	5.9 {4}	0.0004
	Haematocrit (%) ^b^	36.71 ± 5.74	35.51 ± 6.11	37.14 ± 5.55	0.013
	Platelet count (×10^3^/µL) ^a^	232 {114}	221.02 ± 110.63	250.23 ± 92.16	0.002
	INR	1.12 {0.26}	1.17 {0.29}	1.12 {0.17}	0.208
	Glycaemia (mg/dL) ^a^	104 {44}	113 {66}	102 {37}	0.019
	Serum creatinine (mg/dL) ^a^	0.87 {0.39}	1.22 {0.89}	0.8 {0.26}	<0.001
	eGFR (mL/min/1.73 m^2^) ^a^	79.72 {37.11}	52.75 ± 26.04	85.84 ± 27.93	<0.001
	Serum lipase (U/L) ^a^	144.5 {209}	135 {242}	145 {188}	0.291
	Bilirubin (mg/dL) ^a^	4.2 {8.1}	7.1 {11.4}	3.7 {7}	<0.001
	AST (U/L) ^a^	86 {130}	88 {94}	86 {147}	0.538
	ALT (U/L) ^a^	116.5 {175}	101 {100}	132 {216}	0.044
	ALP (U/L) ^a^	285 {343}	397 {482}	243 {279}	<0.001
	Gamma-GT (U/L) ^a^	467 {599}	482 {751}	460 {558}	0.386
	Albumin (g/dL) ^b^	3.03 ± 0.73	2.73 ± 0.71	3.13 ± 0.70	<0.001
	C-reactive protein (mg/dL) ^a^	23.05 {79.4}	48.9 {120}	17.23 {67.8}	<0.001
Sepsis, *n* (%) ^c^		36 (9.1)	21 (20.4)	15 (5.12)	<0.001
Acute cholangitis, n (%) ^c^		223 (56.31)	82 (79.61)	141 (48.12)	<0.001
Indications of ERCP					
	Choledocholithiasis, *n* (%) ^c^	240 (60.6)	72 (69.9)	168 (57.33)	0.026 *
	Malignant biliary obstruction, *n* (%) ^c^	84 (21.21)	21 (20.38)	63 (21.50)	0.889
	Biliary/pancreatic duct stricture, *n* (%) ^c^	72 (18.18)	10 (9.7)	62 (21.16)	0.001 *
Types of ERCP-procedures					
	Biliary/pancreatic drainage, *n* (%) ^c^	240 (60.6)	72 (69.9)	168 (57.33)	0.026 *
	Sphincterotomy, *n* (%) ^c^	84 (21.21)	21 (20.38)	63 (21.50)	0.889
	Biopsy, *n* (%) ^c^	72 (18.18)	10 (9.7)	62 (21.16)	0.001 *
ERCP duration (minutes)		34.9 ± 9.7	33.5 ± 7.3	35.3 ± 10.4	0.189

* Differences are significant; ^a^ variables with non-Gaussian distribution. Results are presented as median and {interquartile range}; ^b^ variables with Gaussian distribution. Results are presented as mean ± standard deviation. *p* was calculated using unpaired Student’s *t*-test; ^c^ results are presented as number (percentage) from the group’s total. *p* was calculated using Fisher’s exact test. AKI—acute kidney injury, ALT—alanine aminotransferase, ALP—alkaline phosphatase, AST—aspartate aminotransferase, CCI—Charlson Comorbidity Index, eGFR—estimated glomerular filtration rate, ERCP—endoscopic retrograde cholangiopancreatography, SIRS—systemic inflammatory response syndrome.

**Table 2 biomedicines-10-03166-t002:** Risk factors for post-ERCP AKI.

Predictor	Unadjusted OR (95% CI)	*p*	Adjusted OR (95% CI)	*p*
Age (per one year)	1.03 (1.01–1.05)	0.004	1.07 (1.02–1.12)	0.001
Choledocholithiasis	1.73 (1.07–2.79)	0.026	3.31 (1.29–8.54)	<0.001
Biliary/pancreatic duct stricture	0.40 (0.19–0.81)	0.012	-	
eGFR (per one mL/min/1.73 m^2^)	0.95 (0.94–0.96)	<0.001	0.91 (0.89–0.97)	<0.001
Acute cholangitis	2.08 (1.39–3.11)	<0.001	-	-
Sepsis	4.74 (2.34–9.62)	<0.001	-	-
C-reactive protein (per one mg/L)	1.01 (1.00–1.01)	<0.001	-	-
Albumin (per one g/dL)	0.45 (0.33–0.63)	<0.001	-	-
Nonrenal CCI (per one point)	1.18 (1.09–1.28)	<0.001	1.21 (1.02–1.44)	0.034
Systemic inflammatory response syndrome	1.89 (1.14–3.16)	0.013	-	
White blood cell count (per unit)	1.07 (1.02–1.12)	0.003	Not used	
Neutrophil counts (per unit)	1.08 (1.03–1.13)	0.001	Not used	
Haematocrit (per unit)	0.95 (0.91–0.99)	0.014	-	
Platelet count (per unit)	0.99 (0.99–1.00)	0.010	1 (1.00–1.01)	0.004
Glycaemia (per one mg/dL)	1.01 (1.00–1.01)	0.012	1.01 (1.00–1.03)	0.021
Bilirubin (per one mg/dL)	1.06 (1.04–1.09)	<0.001	1.12 (1.06–1.18)	<0.001
ALT (per one U/L)	0.99 (0.99–1.00)	0.006	0.97 (0.94–1.00)	0.003
ALP (per one U/L)	1.00 (1.01–1.02)	0.001	1.01 (1.00–1.04)	0.002

ALT—alanine aminotransferase, ALP—alkaline phosphatase, CCI—Charlson Comorbidity Index, eGFR—estimated glomerular filtration rate.

**Table 3 biomedicines-10-03166-t003:** Comparison of death rate, length of hospital stay, and post-ERCP adverse events in patients with and without AKI after ERCP.

Parameter		Total ERCP Group (*n* = 396)	AKI Group (*n* = 103)	Non-AKI Group (*n* = 293)	*p*
Death, *n* (%) ^b^		9 (2.27)	8 (7.76)	1 (0.34)	<0.001
Cause of death, *n* (%)	Infection, *n* (%) ^b^	3 (0.75)	2 (1.94)	1 (0.34)	0.167
	Malignancy progression, *n* (%) ^b^	3 (0.75)	3 (2.91)	0 (0)	0.017
	Heart failure, *n* (%) ^b^	2 (0.5)	2 (1.94)	0 (0)	0.067
	Cardiac arrest of unknown aetiology, *n* (%) ^b^	1 (0.25)	1 (0.97)	0 (0)	0.260
Length of hospital stay, (days) ^a^		5 {5}	7 {6}	5 {5}	0.002
Types of complications					
Pancreatitis, *n* (%) ^b^		14 (3.53)	5 (4.85)	9 (3.07)	0.369
Bleeding, *n* (%) ^b^		2 (0.5)	1 (0.97)	1 (0.34)	0.453
Perforation, *n* (%) ^b^		2 (0.5)	1 (0.97)	1 (0.34)	0.453

^a^ Variables with non-Gaussian distribution. Results are presented as median and {interquartile range}; ^b^ results are presented as number (percentage) from the group’s total. *p* was calculated using Fisher’s exact test. AKI—acute kidney injury, ERCP—endoscopic retrograde cholangiopancreatography.

**Table 4 biomedicines-10-03166-t004:** Univariate and multivariate analysis to determine predictors of in-hospital mortality.

Predictor	Unadjusted OR (95% CI)	*p*	Adjusted OR (95% CI)	*p*
AKI stage 2 and 3	13.94 (3.48–55.77)	<0.001	6.43 (1.48–27.88)	0.013
eGFR (per one mL/min/1.73 m^2^)	0.97 (0.95–0.99)	0.022	0.98 (0.95–1.02)	0.077
Albumin (per one g/dL)	0.19 (0.07–0.49)	0.001	0.18 (0.08–0.39)	<0.001
Nonrenal CCI (per one point)	1.42 (1.15–1.76)	0.001	1.25 (1.03–1.52)	0.025
Systemic inflammatory response syndrome	4.56 (1.20–17.37)	0.026	-	
White blood cell counts (per unit)	1.16 (1.07–1.26)	<0.001	Not used	
Neutrophil counts (per unit)	1.16 (1.06–1.27)	<0.001	Not used	
Bilirubin (per one mg/dL)	1.07 (1.01–1.14)	0.015	-	

AKI—acute kidney injury, CCI—Charlson Comorbidity Index, eGFR—estimated glomerular filtration rate.

## Data Availability

The data that support the findings of this study are available from the corresponding author upon reasonable request.

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
