# Peer review of "Acute Kidney Injury after Endoscopic Retrograde Cholangiopancreatography—A Hospital-Based Prospective Observational Study"

_biomedicines, 2022, doi:10.3390/biomedicines10123166_

Round 1

Reviewer 1 Report (Previous Reviewer 2)

The authors have addressed all my questions. 

Reviewer 2 Report (Previous Reviewer 1)

Dear authors,

/I congratulate you for your work. The manuscript has been considerably improved. Therefore I consider it cand be published in the current form.

This manuscript is a resubmission of an earlier submission. The following is a list of the peer review reports and author responses from that submission.

Round 1

Reviewer 1 Report

Dear authors,

Thank you for the opportunity to provide the peer review of the manuscript. The paper deals with the in cidence and risk factors regarding AKI after ERCP. There are only few studies published regarding this important complication after this procedure. The article is very well and clear written, the results are interestinfg and the conclusions are supported by yhese results. I have only minor observation regarding the reference to the tables' numbers in the text, tehy need to be corrected since some of these references are not ok - see for example line 198 (table 1 is correct, not table 2) and so on.

Author Response

Dear Reviewer,

Thank you for the review. We corrected the mistake: Table 2 was numbered as Table 3 and the previous Table 3 became Table 2.

Reviewer 2 Report

The authors aimed to investigate the incidence of AKI after ERCP and assess potential independent factors associated with the occurrence of AKI post-ERCP and with in-hospital mortality. They found that the incidence of AKI after ERCP reached 26%. The independent factors associated with AKI were: baseline eGFR, non-renal CCI score, choledocholithiasis, and serum bilirubin level at admission. Furthermore, in-hospital mortality rate gradually increased with the rising of AKI severity.

I think It is necessary to show the causes of death in these patients. In addition, it is better to show the presence of proteinuria, bacteremia, sepsis, shock, or DIC. It is interesting to see the correlation between the procedure time of ERCP and the incidence of AKI.

In this paper, they assessed incidence of AKI and in-hospital mortality only in the first 48 hours. It would be better to provide the evaluation of AKI and mortality in the longer term such as 7 or 14 days.

Author Response

Dear Reviewer,

Thank-you for your precious remarks.

In our cohort the in-hospital mortality rate was 2.27%. Of the 9 patients who died, progression of malignancy (3/9; 33.33%) and infections (3/9; 33.33%) were the major causes of death. All other causes of death are presented in table 2.  Deaths associated with the progression of underlying malignancies were more frequent in the AKI group as compared to non-AKI patients (P =0.016). (table 2) (We included these data in the “Results” section, lines 250-255)

Sepsis was present upon admission in 36/396 (9.1%) patients. In the AKI group, 20.39% of patients had sepsis, whereas in non-AKI patients, sepsis was present in 5.12% upon admission. (These data were included in Table 1). Furthermore, we observed that sepsis was associated with the risk for AKI in the univariable regression. However, in the multivariate regression analysis, sepsis was not independently associated with the risk of AKI.  The obtained data are presented in Table 3.

For the present study we did not collect proteinuria, while data about bacteriemia was not available in all patients and did not allow for an accurate statistical analysis.

In our study, duration of ERCP procedure ranged between 10 and 110 minutes, with a mean procedure duration of 34.9 ± 9.7 minutes. (We added this statement in the “Results section” lines 180-182). We did not find a correlation between ERCP procedure duration and incidence of AKI (R2 = 0.004, P = 0.209).

In our cohort, we assessed data concerning the association of AKI with in-hospital mortality. The median length of hospital stay in our group was 5 days. Unfortunately, we could not evaluate mortality after patient’s discharge (i.e. deaths in the community or in another hospital).

Reviewer 3 Report

Dear authors,

I congratulate you for your work!

Concerning indications for ERCP, you stated only three, but since patients with AKI were generally sicker, I am sure there was a group of patients that fulfilled the TG18 criteria for cholangitis (patients both from CDL and obstruction/strictures group); I believe it would be useful to look at this group separately.

To my experience the problem could be even hydration of the patients periprocedurally. As you state in the discussion section, you used rather sensitive definition of AKI. Since the patients undergoing ERCP are around 70 years old and have comorbidities including CHF, in this age group to my opinion even inadequate hydration periprocedurally can result in diagnosing AKI the day after ERCP with this sensitive definition, e.g.  if patient undergoes ERCP let's say at 2 pm, and did not get fluids before the procedure (which I see its not rare), he might increase creatinine hypothetically the next day especially if he has CHF and the physicians were careful with i.v. hydration. Do you provide an information on the hydration of patients periprocedurally? And maybe to discuss this and even deliver a message to the readers that this could be clinically relevant, so they can address this in their clinical practice if needed.

I am surprised with rather low percentage of sphincterotomy performed (51.5%)!

Author Response

Dear Reviewer,

Thank you for your observations.

Preprocedural acute cholangitis, defined according to the 2018 Tokyo Guidelines, was found in 56.31% of patients, the rate of AC being significantly higher in the AKI group versus non-AKI patients (79.61% vs. 48.12%; P<0.001). (Table 1). Causes of AC included choledocholithiasis (63.67%), followed by malignant biliary obstruction (20.62%), and biliary/pancreatic duct strictures (15.69%). (These data were included in the “Results” section, lines 205 -211 and in the Table 1).

Furthermore, we included in the “Material and Methods” section the following sentence: << Acute cholangitis (AC) was defined according to the 2018 Tokyo Guidelines, and the diagnostic criteria included systemic inflammation, cholestasis, and imaging findings>> with this reference - Kiriyama S, Kozaka K, Takada T, Strasberg SM, Pitt HA, Gabata T, Hata J, Liau KH, Miura F, Horiguchi A, Liu KH, Su CH, Wada K, Jagannath P, Itoi T, Gouma DJ, Mori Y, Mukai S, Giménez ME, Huang WS, Kim MH, Okamoto K, Belli G, Dervenis C, Chan ACW, Lau WY, Endo I, Gomi H, Yoshida M, Mayumi T, Baron TH, de Santibañes E, Teoh AYB, Hwang TL, Ker CG, Chen MF, Han HS, Yoon YS, Choi IS, Yoon DS, Higuchi R, Kitano S, Inomata M, Deziel DJ, Jonas E, Hirata K, Sumiyama Y, Inui K, Yamamoto M. Tokyo Guidelines 2018: diagnostic criteria and severity grading of acute cholangitis (with videos). J Hepatobiliary Pancreat Sci. 2018;25:17–30, doi: 10.1002/jhbp.512

We evaluated the association between AC and AKI. In the univariate regression analysis, AC was associated with risk for AKI development with an OR = 2.08, 95%CI: 1.39 to 3.11. (we included this data in Table 2).  In the next step, we performed a multivariate analysis in which we added acute cholangitis as a distinct variable. However, in the multivariate regression analysis, AC was not independently associated with the risk of AKI. This finding could be due to the fact that we analyzed the association of AC with the risk of AKI and we did not stratify AC by disease severity. According to the literature it seems that only TG grade III is an independent predictor for development of AKI in patients with AC. [Lee TW, Bae W, Kim S, Choi J, Bae E, Jang HN, Chang SH, Park DJ. Incidence, risk factors, and prognosis of acute kidney injury in hospitalized patients with acute cholangitis. PLoS One. 2022 Apr 14;17(4):e0267023. doi: 10.1371/journal.pone.0267023].

Because, patients subjected to ERCP are fasting, in our department these patients receive on a routine basis periprocedural hydration with normal intravenous saline with a maximum of 1.5 mL/kg per h and 3 L per 24 h. However, the hydration measures are adapted to the patient’s comorbidities (i.e. congestive heart failure) in order to avoid overhydration-related complications, such as pulmonary edema and congestive heart failure.These sentences were added in the “Materials and Methods” section, lines 131-136.

Moreover, as you suggested we added in the “Discussion” section, lines 392-395, the following statement : << The results of our study highlight the fact that patients undergoing ERCP are vulnerable to development of AKI. For these patients it is very important to intervene early and to provide an adequate periprocedural hydration, while avoiding fluid overload, and also to withdraw or not to add possible nephrotoxins.>>